# The Design of State-Dependent Switching Rules for Second-Order Switched Linear Systems Revisited

Xiaolan Yuan and Yusheng Zhou *



School of Mathematics and Statistics, Guizhou University, Guiyang 550025, China
* Correspondence: yszhou@gzu.edu.cn

**Abstract:** This paper focuses on the asymptotic stability of second-order switched linear systems with positive real part conjugate complex roots for each subsystem. Compared with available studies, a more appropriate state-dependent switching rule is designed to stabilize a switched system with the phase trajectories of two subsystems rotating outward in the same direction or the opposite direction. Finally, several numerical examples are used to illustrate the effectiveness and superiority of the proposed method.

**Keywords:** switched system; unstable subsystem; asymptotic stability; state-dependent switching rule





## 1. Introduction

For switched systems, differential equations are used to describe the dynamic behavior of the continuous characteristic, which are marked as subsystems. Meanwhile, a piecewise constant function is adopted to describe the discrete characteristic, which is referred to as a switching rule. As an important part of hybrid systems, switched systems have been exhibited in many practical fields, such as engine systems [1], network systems [2], stirred reactor systems [3], mobile robot systems [4], etc. Moreover, the expected performance of some complex controlled systems can be efficiently achieved by switching control strategies [5,6]. Taking the modern variable-speed wind turbine for example, it can switch back and forth between low and high wind speed modes according to the current wind speed, thus capturing as much wind energy as possible at rated power [7]. Meanwhile, considerable attention has been paid to investigations of switched systems over the last decades. Wu et al. [8] studied the stability of stochastic switched systems via probabilistic analysis. Tian et al. [9] considered the controllability and observability of multi-agent switched systems with continuous and discrete subsystems. Liu et al. [10] and Niu et al. [11,12] designed neural adaptive control strategies for nonlinear switched systems. However, in a practical controlled system, actuator or sensor failure would inevitably occur, thus destabilizing the original stable subsystem. Therefore, it would be very meaningful to conduct studies on switched systems with unstable subsystems, especially for switched systems with fully unstable subsystems, which in recent years has attracted the attention of many researchers and led to the development of numerous significant results. For instance, a fault detection observer was designed to address subsystem instability caused by unobservable factors [13]. A sufficient asymptotic stability condition was presented by a time-dependent strategy for a state-constrained switched system with multiple unstable subsystems [14]. However, to the best of the authors' knowledge, a switched system with the characteristic roots of each subsystem being positive has been rarely considered so far, and is thus the research subject of this paper.

The succesful design of a switching rule to render a switched system asymptotically stable presents an interesting problem that has attracted the attention of many scholars. As

a robust closed-loop switching mechanism, state-dependent switching is relatively suitable for solving the above problem. Wu et al. [15] discussed the stability problem of stochastic switched systems via state-dependent switching rule. Liu and Long [16] designed a state-dependent switching rule with guaranteed dwell time to stabilize a class of nonlinear switched systems, with the help of a sum-of-squares constraint approach and an improved path-tracing method. Guo et al. [17] considered the multi-stability of switched neural networks with sinusoidal activation functions under state-dependent switching. Yang and Li [18] achieved the stability of stochastic switched neural networks with time-varying parameter uncertainty through a state-dependent switching approach. For a switched system where the characteristic roots of each subsystem are all positive real parts, the state-dependent switching rule would also be very applicable.

In practical problems, all subsystems may be unstable due to extreme operating environments, measurement failures, or actuator failures. In general, it could be difficult to make the switched system with fully unstable subsystems asymptotically stable via time-dependent switching rules. In particular, when the characteristic roots of each subsystem are all positive real parts, there are no stable factors in each subsystem. In this situation, it is impossible to compensate for the divergence of unstable subsystems through the stability factors of subsystems. That is, the time-dependent switching rule does not work, and only state-dependent switching rule can be applied to achieve asymptotic stability of such switched systems. In fact, for switched systems with the characteristic roots of each subsystem being all positive real parts, there has been little relevant research work, and no systematic universal method has been formed at present. To the best of the authors' knowledge, in the existing representative research literature, Pettersson investigated a special class of linear switched systems, in which the characteristic roots of each subsystem were all positive real parts [19]. A state-dependent switching rule was constructed by the largest region function strategy to achieve system asymptotic stability. However, the obtained results are somewhat conservative for adopting a linear matrix inequality approach. Furthermore, the calculation process is relatively complex. In order to fundamentally address the drawback of poor conservativeness, Ref. [20] constructed an energy function with practical physical meaning for each subsystem by introducing an invertible transformation. By analyzing the energy ratio functions of two subsystems, two switching lines with maximum and minimum energy loss were obtained to design a proper state-dependent switching rule. The obtained result is evidently simpler and less conservative than [19], and also demonstrates faster convergence of system states.

It is worth noting that the switching rule proposed in [20] is only applicable to switched systems where the phase trajectories of two subsystems rotate counter-clockwise. When the phase trajectories of both subsystems rotate clockwise, or one rotates counter-clockwise while the other clockwise, such a switching rule cannot guarantee system stability. Based on the above-mentioned discussions, two improved state-dependent switching rules are put forward in this paper. The main innovations are as follows:

(i) A more suitable state-dependent switching rule is designed to stabilize a switched system with the phase trajectories of both subsystems rotating in the same direction.

(ii) When the phase trajectories of two subsystems rotate outward in opposite directions, a novel state-dependent switching rule is proposed to guarantee system stability, by judging whether the system state satisfies a critical switching condition or not.

In addition to the introduction in Section 1, the remainder of the paper is organized as follows. Section 2 presents the knowledge of the second-order linear switched system. An improved switching rule is designed in Section 3 for the switched system with phase trajectories of subsystems rotating in the same direction. Section 4 designs a novel switching rule for switched systems in which the phase trajectories of subsystems rotate in opposite directions. Subsequently, two examples are given in Section 5 to illustrate the effectiveness of the proposed method. Finally, some conclusions of the paper are presented in Section 6.

## 2. System Description and Preliminaries

In general, the mathematical model of a linear switched system is described by

$$\begin{cases} \dot{\mathbf{x}}(t) = \mathbf{A}_{\sigma(t)}\mathbf{x}(t), \\ \mathbf{x}(0) = \mathbf{x_0}. \end{cases} \tag{1}$$

Here, $\sigma(t) : [0, \infty) \to \Lambda = \{1, 2, 3...N\}$ is a piecewise function that represents a switching rule. $N$ is the number of subsystems in the switched system. $\mathbf{x}(t) \in \mathbb{R}^n$ denotes the state vector and $\mathbf{x_0}$ stands for the given initial value of the system. $\mathbf{A}_i \in \mathbb{R}^{n \times n}$, $i \in \Lambda$ is the state matrix of the $i$th subsystem. In this paper, a second-order switched system with two linear subsystems is considered, in which the eigenroots of each subsystem matrix are a pair of positive real part conjugate complex roots. That is to say, $N = 2$, $\mathbf{x}(t) \in \mathbb{R}^2$ and $\mathbf{A}_i \in \mathbb{R}^{2 \times 2}$. Furthermore, $\mathbf{x}(t)$ and $\mathbf{A}_i$ are specifically depicted as

$$\mathbf{x}(t) = \begin{pmatrix} x_1(t) \\ x_2(t) \end{pmatrix}, \quad \mathbf{A}_i = \begin{pmatrix} a_i & b_i \\ c_i & d_i \end{pmatrix}, \quad i = 1, 2.$$

To elaborate the subsequent work in more clarity, some definitions, hypothesis and lemma should be distinctly provided.

**Definition 1** ([21]). *Under the switching rule $\sigma(t)$, if the solution $\mathbf{x}(t)$ of the switched system (1) is bounded for all $t \in [0, \infty)$ and $\lim_{t \to \infty} \mathbf{x}(t) = \mathbf{0}$ holds for equilibrium point $\mathbf{x}_e = \mathbf{0}$, then the switched system (1) is said to be asymptotically stable at equilibrium point $\mathbf{x}_e = \mathbf{0}$.*

**Definition 2** ([20]). *Consider the following second-order linear system*

$$\begin{pmatrix} \dot{x}_1 \\ \dot{x}_2 \end{pmatrix} = \begin{pmatrix} a & b \\ c & d \end{pmatrix} \begin{pmatrix} x_1 \\ x_2 \end{pmatrix}, \quad c \neq 0, \tag{2}$$

*where its characteristic roots are a pair of conjugate complex roots with positive real parts. Then, the energy function presented by the sum of kinetic and potential energy, is defined as*

$$E = \frac{1}{2}(ad - bc)x_1^2 + \frac{1}{2}(ax_1 + bx_2)^2. \tag{3}$$

*Here, $ad - bc > 0$ denotes the equivalent stiffness coefficient and $a + d > 0$ stands for the equivalent damping coefficient.*

**Hypothesis 1.** *Without loss of generality, $a_1d_1 - b_1c_1 > a_2d_2 - b_2c_2$ is assumed. Namely, the equivalent stiffness coefficient of the first subsystem is greater than that of the second subsystem.*

**Lemma 1** ([20]). *When the phase trajectories of both second-order subsystems given in (1) rotate counter-clockwise, a state-dependent switching rule of the switched system (1) can be designed as*

$$\sigma(t) = \begin{cases} 1, & (x_2(t) - k_1x_1(t)) \cdot (x_2(t) - k_2x_1(t)) \geq 0, \\ 2, & (x_2(t) - k_1x_1(t)) \cdot (x_2(t) - k_2x_1(t)) < 0. \end{cases} \tag{4}$$

*Here, $k_1$, $k_2$ are two constants with $k_1 < k_2$ and $x_2 = k_1x_1$, $x_2 = k_2x_1$ make the following energy ratio function*

$$\frac{E_1}{E_2} = \frac{\frac{1}{2}(a_1d_1 - b_1c_1) + \frac{1}{2}(a_1 + b_1k)^2}{\frac{1}{2}(a_2d_2 - b_2c_2) + \frac{1}{2}(a_2 + b_2k)^2}, \quad k = \frac{x_2}{x_1} \tag{5}$$

*take the maximum and minimum, respectively.*

**Remark 1.** *According to Lemma 1, by using the switching rule (4), the energy loss of the switched system (1) is the largest in a switching loop, so that the system state can converge to the equilibrium point as fast as possible.*

For the switched system (1) with characteristic roots of each subsystem being conjugate complex roots with positive real parts, the switching rule (4) is quite effective. However, ref. [20] only investigates the stability of a switched system with phase trajectories of both subsystems rotating counter-clockwise. The cases of two phase trajectories rotating clockwise or one rotating clockwise while the other rotates counter-clockwise are not considered. Motivated by the above discussion, the objectives of this paper are two-fold.

(i) One is to propose an improved state-dependent switching rule for stabilizing the switched system with the phase trajectories of two subsystems rotating outward in the same direction.

(ii) The other is to design an appropriate state-dependent switching rule to guarantee the asymptotic stability of the switched system with phase trajectories of two subsystems rotating in opposite directions.

## 3. The Case of Two Subsystems with Same Rotation Direction of Phase Trajectories

To address the stability problem of a switched system with phase trajectories of two subsystems rotating in the same direction, we first need to determine the rotation direction of phase trajectory for a second-order linear system. Thus, the following lemma is necessary.

**Lemma 2.** *The phase trajectory of system (2) rotates clockwise if and only if $b - \frac{ad}{c} > 0$. Furthermore, the phase trajectory of system (2) rotates counter-clockwise if and only if $b - \frac{ad}{c} < 0$.*

**Proof of Lemma 2.** We first show the sufficient proof. Since the characteristic roots of system (2) are positive real conjugate complex roots, the phase trajectory is spirally divergent and $c \neq 0$. Then, set a point on the phase trajectory as $(x_1, x_2)^{\mathrm{T}} = (-\frac{d}{c}, 1)^{\mathrm{T}}$. Substituting it into system (2) yields

$$\dot{x}_1 = b - \frac{ad}{c}, \quad \dot{x}_2 = 0.$$

When $b - \frac{ad}{c} > 0$, the tangent vector of phase trajectory is horizontal to the right. This implies that the phase trajectory rotates clockwise. Similarly, the tangent vector is horizontal to the left if $b - \frac{ad}{c} < 0$. Accordingly, the phase trajectory rotates counter-clockwise.

In the next step, we exhibit the necessary proof. Obviously, when the phase trajectory of system (2) rotates clockwise, the tangent vector at point $(x_1, x_2)^{\mathrm{T}}$ is horizontal to the right. This implies $b - \frac{ad}{c} > 0$. Conversely, when the phase trajectory of the system rotates counter-clockwise, the tangent vector at point $(x_1, x_2)^{\mathrm{T}}$ is horizontal to the left. Accordingly, it indicates $b - \frac{ad}{c} < 0$. This completes the proof. □

In fact, in order to stabilize the switched system (1) with same rotation direction of phase trajectories, we need to switch from the first (second) subsystem to the second (first) subsystem at the maximum (or minimum) value of energy ratio function $\frac{E_1}{E_2}$. Therefore, the increased energy from operation unstable subsystems can be compensated by the decreased energy from system switching. On this basis, an improved state-dependent switching rule is proposed as follows

$$\sigma(t) = \begin{cases} 1, & \phi \cdot (x_2(t) - k_1 x_1(t)) \cdot (x_2(t) - k_2 x_1(t)) \geq 0, \\ 2, & \phi \cdot (x_2(t) - k_1 x_1(t)) \cdot (x_2(t) - k_2 x_1(t)) < 0. \end{cases} \tag{6}$$

Here, the slope of $x_1 = 0$ is assumed to be negative infinity. $\phi$ is a sign function, described by

$$\phi = \begin{cases} 1, & k_1 - k_2 > 0 \text{ and } b_i - \frac{a_i d_i}{c_i} > 0, \\ -1, & k_1 - k_2 < 0 \text{ and } b_i - \frac{a_i d_i}{c_i} > 0, \\ -1, & k_1 - k_2 > 0 \text{ and } b_i - \frac{a_i d_i}{c_i} < 0, \\ 1, & k_1 - k_2 < 0 \text{ and } b_i - \frac{a_i d_i}{c_i} < 0. \end{cases} \tag{7}$$

The switching rule (6) includes rule (4) and is also applicable to the switched system with the phase trajectories of two subsystems rotating clockwise. That is, when two subsystems rotate in the same direction, either clockwise or counter-clockwise, the switching rule (6) can be adopted to guarantee the asymptotic stability of the switched system.

In order to better understand the above switching rule, we further present its geometric meaning. As shown in Figure 1a,d, when the system state goes from region $Q_1$ to $Q_2$ through the critical line $x_2 = k_1 x_1$, and the system switches from the first subsystem to the second subsystem, then the energy loss would be maximized under Hypothesis 1. On the contrary, when the system state goes from region $Q_2$ to $Q_1$ through the critical line $x_2 = k_2 x_1$, and the system switches from the second subsystem to the first subsystem, then the increased energy would be minimized. Thus, the first subsystem should be run in $(x_2 - k_1 x_1) \cdot (x_2 - k_2 x_1) \geq 0$ and the second subsystem should be run in $(x_2 - k_1 x_1) \cdot (x_2 - k_2 x_1) < 0$. Likewise, as reflected in Figure 1b,c, if the second subsystem is activated at $(x_2 - k_1 x_1) \cdot (x_2 - k_2 x_1) > 0$ and the first subsystem is activated at $(x_2 - k_1 x_1) \cdot (x_2 - k_2 x_1) \leq 0$, the total energy loss in the switching loop would be maximized. Based on the above analysis, the switching rule (6) is suitable for stabilizing the switched system (1) in which the phase trajectories of two subsystems rotate in the same direction.

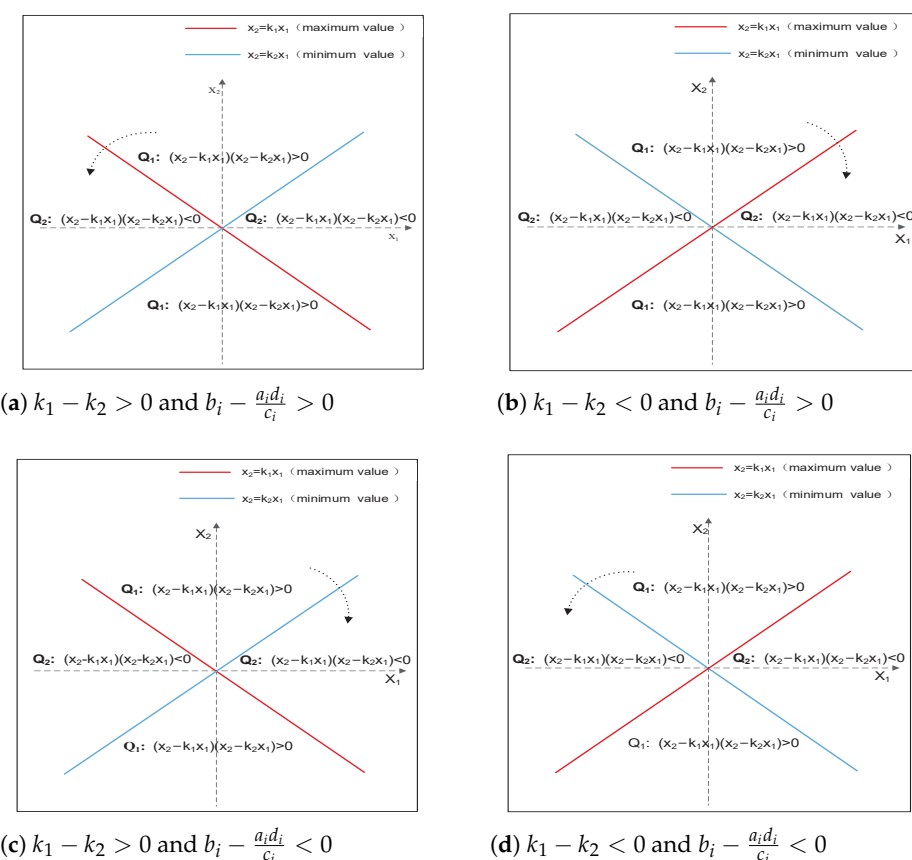

(a) $k_1 - k_2 > 0$ and $b_i - \frac{a_i d_i}{c_i} > 0$      (b) $k_1 - k_2 < 0$ and $b_i - \frac{a_i d_i}{c_i} > 0$

(c) $k_1 - k_2 > 0$ and $b_i - \frac{a_i d_i}{c_i} < 0$      (d) $k_1 - k_2 < 0$ and $b_i - \frac{a_i d_i}{c_i} < 0$

**Figure 1.** A diagram of system switching for different cases given in (7).

## 4. The Case of Two Subsystems with Opposite Rotation Directions of Phase Trajectories

In this section, we investigate the stability of switched systems with phase trajectories of two subsystems rotating outwards in opposite directions.

By means of the switching rule (6), the phase plane is divided into four switching regions, and only one subsystem can be operated in each switching region. When the phase trajectories of two subsystems rotate in opposite directions, the system state may switch back and forth on one of the two critical switching lines if switching rule (6) is adopted. In order to be more specific, an example is presented as below.

**Example 1.** *Consider a switched system consisting of the following two subsystem state matrices*

$$\mathbf{A}_1 = \begin{pmatrix} 1 & 4 \\ -80 & 2 \end{pmatrix}, \quad \mathbf{A}_2 = \begin{pmatrix} \frac{1}{2} & -6 \\ 10 & \frac{1}{2} \end{pmatrix}, \tag{8}$$

*with the initial condition being* $\mathbf{x}(0) = (-1, 2)^{\mathrm{T}}$.

By calculation, the eigenvalues of the first subsystem and the second subsystem are, respectively, given by $1.50 \pm 17.88i$ and $0.50 \pm 7.75i$. Since the characteristic roots of both subsystems are positive real parts, the two subsystems are unstable. According to Lemma 2, the phase trajectory of the first subsystem rotates clockwise, while the second one rotates counter-clockwise, as depicted in Figure 2. Clearly, the phase trajectories of the two subsystems rotate in opposite directions.

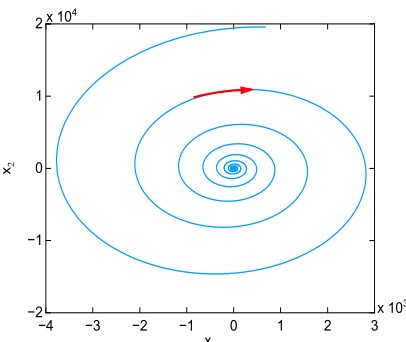

(**a**) The phase trajectory of the first subsystem

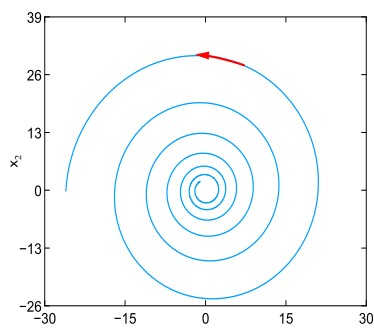

(**b**) The phase trajectory of the second subsystem

**Figure 2.** The phase trajectories of the two subsystems.

In view of Equation (5), the energy ratio function between the first and second subsystems is obtained as

$$\frac{E_1}{E_2} = \frac{161 + \frac{(1+4k)^2}{2}}{\frac{241}{8} + \frac{(\frac{1}{2} - 6k)^2}{2}}. \tag{9}$$

Taking the derivative of Equation (9) with respect to $k$ yields the following two critical switching lines

$$x_2 = k_1 x_1 = 0.11 x_1, \quad x_2 = k_2 x_1 = -55.63 x_1. \tag{10}$$

Here, $x_2 = 0.11 x_1$ and $x_2 = -55.63 x_1$ make the energy ratio function (9) take the maximum and minimum values, respectively. As depicted in Figure 3, the switching rule (6) cannot stabilize the switched system (8) irrespective of whether $\phi = 1$ or $\phi = -1$. In particular, the system state switches back and forth on the critical switching line $x_2 = 0.11 x_1$ when $\phi = 1$, as shown in Figure 4a. Similarly, in Figure 4b, the system state hovers around the critical switching line $x_2 = -55.63 x_1$ when $\phi = -1$.

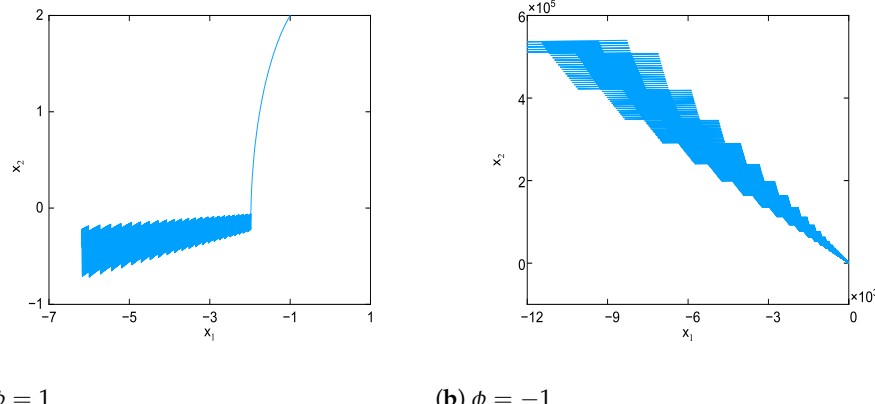

**(a)** $\phi = 1$          **(b)** $\phi = -1$

**Figure 3.** The phase diagrams of system (8) under the switching rule (6).

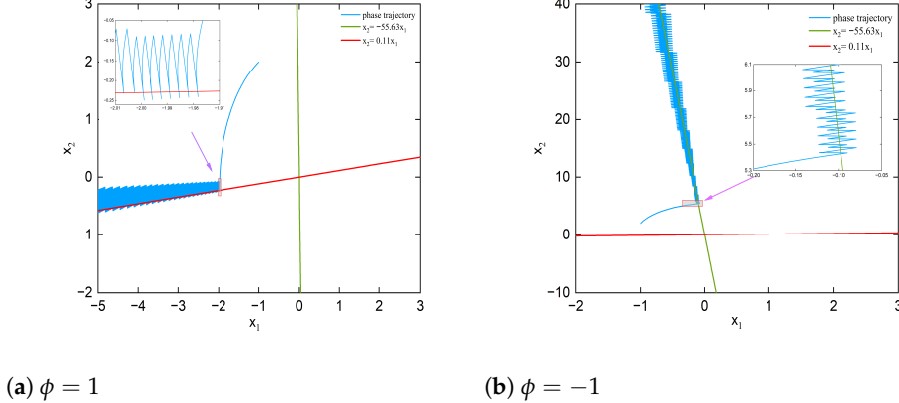

**(a)** $\phi = 1$          **(b)** $\phi = -1$

**Figure 4.** Phase diagrams and critical switching lines.

We can see from the above simulation that the switched system fails to switch from the first subsystem to the second subsystem at the maximum value of $\frac{E_1}{E_2}$, or from the second subsystem back to the first subsystem at the minimum value of $\frac{E_1}{E_2}$. In this situation, the energy reduced by system switching cannot offset the energy increased from the unstable subsystem operation. Naturally, the switching rule (6) cannot stabilize the switched system with phase trajectories of two subsystems rotating in opposite directions. In order to avoid the system switching occurring back and forth on one of the two critical switching lines, the switching rule (6) is improved as follows

$$
\sigma(t + \triangle t) = \begin{cases} 1, & \sigma(t) = 1 \text{ and } (x_2(t) - k_1 x_1(t)) \cdot (x_2(t - \triangle t) - k_1 x_1(t - \triangle t)) > 0, \\ 2, & \sigma(t) = 1 \text{ and } (x_2(t) - k_1 x_1(t)) \cdot (x_2(t - \triangle t) - k_1 x_1(t - \triangle t)) \leq 0, \\ 1, & \sigma(t) = 2 \text{ and } (x_2(t) - k_2 x_1(t)) \cdot (x_2(t - \triangle t) - k_2 x_1(t - \triangle t)) \leq 0, \\ 2, & \sigma(t) = 2 \text{ and } (x_2(t) - k_2 x_1(t)) \cdot (x_2(t - \triangle t) - k_2 x_1(t - \triangle t)) > 0. \end{cases} \tag{11}
$$

Here, $\triangle t$ is the step length of time in the numerical calculation, which is defined as 0.001 seconds in this paper.

In what follows, we will present a detailed explanation of the switching rule (11). For simplicity, we assume that the first subsystem is activated at the initial time, and the equivalent stiffness coefficients of two subsystems satisfy Hypothesis 1. In addition, $k_1$ and $k_2$ are assumed to be the maximum and minimum points of $\frac{E_1}{E_2}$, respectively. To avoid switching back and forth on one critical switching line only, we need to make a restriction on the switching rule (6). That is, the system is required to switch from the first (or second) subsystem to the second (or first) one if and only if the system state

crosses the switching line $x_2 = k_1 x_1$ (or $x_2 = k_2 x_1$). Moreover, whether the system state crosses the switching line $x_2 = k_1 x_1$ can be judged by considering the inequation $(x_2(t) - k_1 x_1(t)) \cdot (x_2(t - \triangle t) - k_1 x_1(t - \triangle t)) > 0$. In other words, the system state crosses the switching line $x_2 = k_1 x_1$ if and only if $(x_2(t) - k_1 x_1(t)) \cdot (x_2(t - \triangle t) - k_1 x_1(t - \triangle t)) < 0$ holds. Based on the above restriction, we can obtain the switching rule (11) and the system switching process is as follows. Starting from the first subsystem, we need to determine whether the inequation $(x_2(t) - k_1 x_1(t)) \cdot (x_2(t - \triangle t) - k_1 x_1(t - \triangle t)) > 0$ holds from time to time during its operation. If so, the first subsystem continues to run. Otherwise, the system state would cross the switching line $x_2 = k_1 x_1$, and then the system needs to be switched from the first subsystem to the second subsystem. Similarly, it is necessary to judge whether the inequation $(x_2(t) - k_2 x_1(t)) \cdot (x_2(t - \triangle t) - k_2 x_1(t - \triangle t)) > 0$ holds in the operation of the second subsystem. Repeating the above switching steps, the energy loss from system switching in a switching loop could be maximized as much as possible, thereby rapidly achieving asymptotic stability.

**Remark 2.** *Although the switching mechanism proposed in this manuscript is suitable for a second-order switched system with three subsystems, the switching sequence of subsystems bears a significant impact on system stability. As such, constructing an optimal switching sequence that enables rapid convergence of the system presents a challenging problem under the proposed state-dependent switching rule.*

### 5. Simulation Results

**Example 2.** *To illustrate the effectiveness of the switching rule (6), we consider a switched system with the following two state matrices*

$$\mathbf{A}_1 = \begin{pmatrix} \frac{1}{3} & -10 \\ 100 & \frac{1}{3} \end{pmatrix}, \quad \mathbf{A}_2 = \begin{pmatrix} 1 & -3 \\ 2 & \frac{1}{2} \end{pmatrix}, \tag{12}$$

*where the initial condition is* $\mathbf{x}(0) = (1, -1)^{\mathrm{T}}$.

By calculation, the eigenvalues of matrix $\mathbf{A}_1$ are $0.33 \pm 31.62i$ and the eigenvalues of matrix $\mathbf{A}_2$ are $0.75 \pm 2.44i$. It implies that both subsystems are unstable. Applying Lemma 2, phase trajectories of both subsystems rotate counter-clockwise. As can be seen in Figure 5, the phase trajectories of two subsystems rotate in the same direction, which indicates that the switching rule (6) is effective.

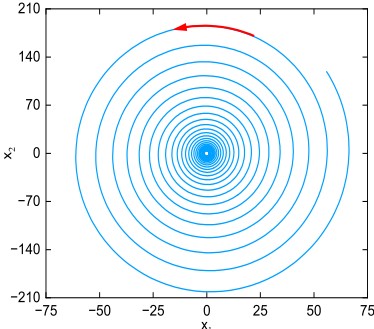 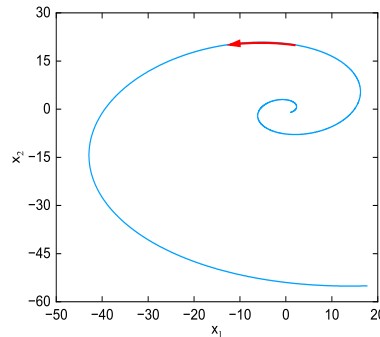

(**a**) The phase trajectory of the first subsystem    (**b**) The phase trajectory of the second subsystem

**Figure 5.** The phase trajectories of the two subsystems.

According to Equation (5), the energy ratio function of two subsystems is calculated as

$$\frac{E_1}{E_2} = \frac{\frac{9001}{18} + \frac{(\frac{1}{3} - 10k)^2}{2}}{\frac{13}{4} + \frac{(1 - 3k)^2}{2}}. \tag{13}$$

By solving $\frac{d(\frac{E_1}{E_2})}{dk} = 0$, two critical switching lines are obtained as

$$x_2 = k_1 x_1 = 0.36 x_1, \quad x_2 = k_2 x_1 = -30.92 x_1. \tag{14}$$

Here, $x_2 = 0.36 x_1$ and $x_2 = -30.92 x_1$ make the energy ratio function (13) take the maximum and minimum values, respectively. For the two subsystems, since the rotation directions of their phase trajectories are counter-clockwise, $b_i - \frac{a_i d_i}{c_i} < 0$. Therefore, the switching rule of switched system (12) is finally derived as

$$\sigma(t) = \begin{cases} 1, & -(x_2(t) - 0.36x_1(t)) \cdot (x_2(t) + 30.92x_1(t)) \geq 0, \\ 2, & -(x_2(t) - 0.36x_1(t)) \cdot (x_2(t) + 30.92x_1(t)) < 0. \end{cases}$$

From Figure 6, we can see that the switched system (12) can be quickly stabilized to the equilibrium point under the switching rule (6). That is, the switching rule (6) can stabilize the switched system (1) with the phase trajectories of two subsystems rotating in the same direction.

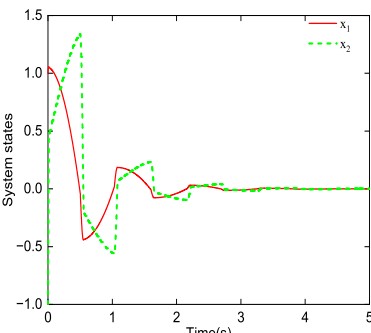
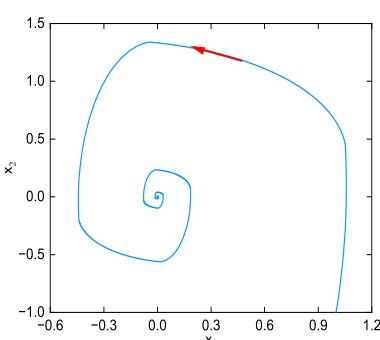

(**a**) The time history of system states    (**b**) The phase diagram of switched system (12)

**Figure 6.** The switched system (12) is asymptotically stable under the switching rule (6).

**Example 3.** *The switched system* (8) *in Example 1 is used to clarify the validity of the switching rule* (11).

Since the phase trajectories of two subsystems rotate in opposite directions, the switching rule (11) is employed. Substituting Equation (10) into this state-dependent switching rule, one obtains

$$\sigma(t + \triangle t) = \begin{cases} 1, & \sigma(t) = 1 \text{ and } (x_2(t) - 0.11x_1(t)) \cdot (x_2(t - \triangle t) - 0.11x_1(t - \triangle t)) > 0, \\ 2, & \sigma(t) = 1 \text{ and } (x_2(t) - 0.11x_1(t)) \cdot (x_2(t - \triangle t) - 0.11x_1(t - \triangle t)) \leq 0, \\ 1, & \sigma(t) = 2 \text{ and } (x_2(t) + 55.63x_1(t)) \cdot (x_2(t - \triangle t) + 55.63x_1(t - \triangle t)) \leq 0, \\ 2, & \sigma(t) = 2 \text{ and } (x_2(t) + 55.63x_1(t)) \cdot (x_2(t - \triangle t) + 55.63x_1(t - \triangle t)) > 0. \end{cases}$$

As can be seen in Figure 7, the switching rule (11) makes the switched system (8) converge to the equilibrium point quickly. This indicates that the switching rule (11) is very effective when the phase trajectories of the two subsystems rotate in opposite directions.

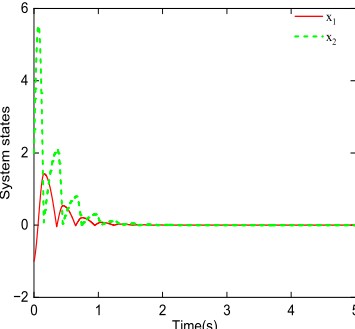 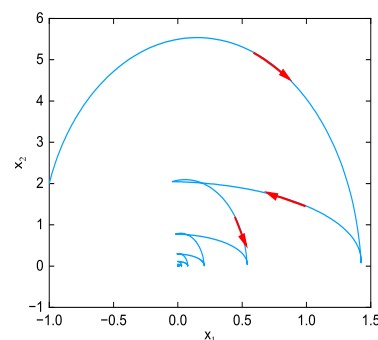

(**a**) The time histories of system states  (**b**) The phase diagram of switched system (8)

**Figure 7.** The switched system (8) is asymptotically stable under the switching rule (11).

**Example 4.** *Consider a switched system* (1) *with the following three state matrices*

$$\mathbf{A}_1 = \begin{pmatrix} 1 & 100 \\ -100 & 1 \end{pmatrix}, \quad \mathbf{A}_2 = \begin{pmatrix} \frac{1}{3} & -10 \\ 30 & \frac{1}{2} \end{pmatrix}, \quad \mathbf{A}_3 = \begin{pmatrix} \frac{1}{4} & 1 \\ -9 & \frac{1}{4} \end{pmatrix}, \tag{15}$$

*where the initial condition is* $\mathbf{x}(0) = (2,4)^{\mathrm{T}}$.

The eigenvalues of matrices $\mathbf{A}_1$, $\mathbf{A}_2$, and $\mathbf{A}_3$ are calculated as $1 \pm 100i$, $0.42 \pm 17.32i$, and $0.25 + 3i$, respectively. Clearly, all subsystems are completely unstable. As shown in Figure 8, the phase trajectories of the first and third subsystems rotate clockwise outside, while the phase trajectory of the second subsystem rotates counter-clockwise outside.

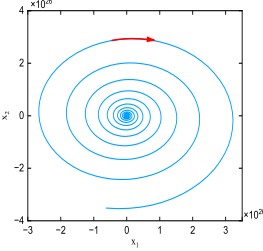 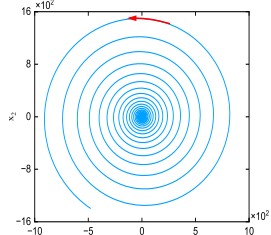 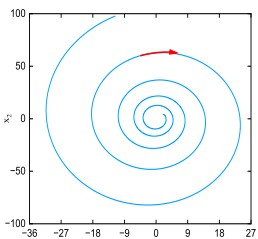

(**a**) The phase trajectory of the first subsystem  (**b**) The phase trajectory of the second subsystem  (**c**) The phase trajectory of the third subsystem

**Figure 8.** The phase trajectories of the three subsystems.

For a switched system (15) with three subsystems, there are six different switching sequences. Without loss of generality, the switching sequence is assumed to be $(i_1 \rightarrow i_2 \rightarrow i_3 \rightarrow i_4)$, where $i_4 = i_1$ and $i_s \in \{1, 2, 3\}$, $(s = 1, 2, 3)$ indicates that the $i_s$th subsystem is running.

If the phase trajectories of the $i_s$th subsystem and the $i_{s+1}$th subsystem rotate in the same direction, the switching rule (6) is adopted. Since the $i_{s+1}$th subsystem does not need to be switched back to the $i_s$th subsystem, the switching rule should be improved by adding the limited function $\sigma(t) = i_s$. As such, the switching rule is rewritten as

$$\sigma(t + \triangle t) = \begin{cases} i_s, & \sigma(t) = i_s \text{ and } \phi \cdot (x_2(t) - k_1 x_1(t)) \cdot (x_2(t) - k_2 x_1(t)) \geq 0, \\ i_{s+1}, & \sigma(t) = i_s \text{ and } \phi \cdot (x_2(t) - k_1 x_1(t)) \cdot (x_2(t) - k_2 x_1(t)) < 0. \end{cases}$$

Here, $x_2 = k_1 x_1$ and $x_2 = k_2 x_1$ make the energy ratio function $\frac{E_{i_s}}{E_{i_{s+1}}}$ take the maximum and minimum values, respectively. If the phase trajectories of the $i_s$th and the $i_{s+1}$th

subsystems rotate in opposite directions, the switching rule (11) is used. Similarly, due to the unidirectional switching, the switching rule is formulated by

$$\sigma(t + \triangle t) = \begin{cases} i_s, & \sigma(t) = i_s \text{ and } (x_2(t) - k_1 x_1(t)) \cdot (x_2(t - \triangle t) - k_1 x_1(t - \triangle t)) > 0, \\ i_{s+1}, & \sigma(t) = i_s \text{ and } (x_2(t) - k_1 x_1(t)) \cdot (x_2(t - \triangle t) - k_1 x_1(t - \triangle t)) \le 0. \end{cases}$$

Here, $x_2 = k_1 x_1$ is the maximum point of the energy ratio function $\frac{E_{i_s}}{E_{i_{s+1}}}$.

Based on the above analysis, we only use switching sequences $(1\to3\to2\to1)$ and $(1\to2\to3\to1)$ as examples to illustrate the effect of switching sequences on system stability. As can be seen in Figure 9, the switched system (15) rapidly converges to the equilibrium point under the switching sequence $(1\to3\to2\to1)$. However, the switched system (15) diverges outward under the switching sequence $(1\to2\to3\to1)$, as depicted in Figure 10. From Figures 9 and 10, we can conclude that the switching sequence plays a crucial role in system stability. Moreover, constructing an appropriate switching sequence is both difficult and important for the switched system (1) with $N \ge 3$.

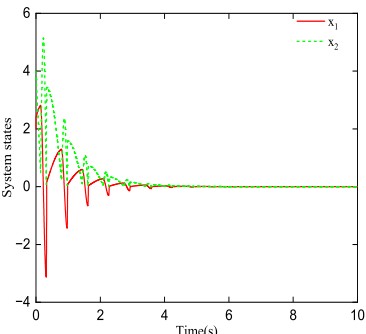
(**a**) The time histories of system states

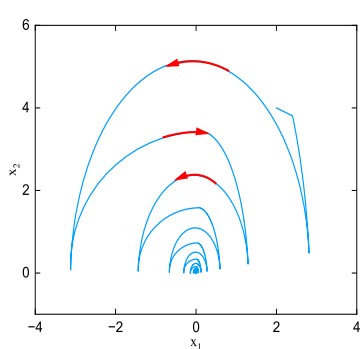
(**b**) The phase diagram of switched system (15)

**Figure 9.** The switched system (15) is asymptotically stable under the switching rule $(1\to3\to2\to1)$.

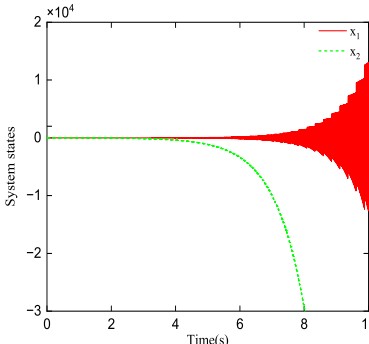
(**a**) The time histories of system states

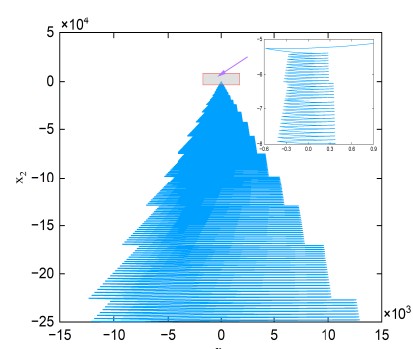
(**b**) The phase diagram of switched system (15)

**Figure 10.** The switched system (15) is not stable under the switching rule $(1\to2\to3\to1)$.

## 6. Conclusions

The asymptotic stability of a class of second-order switched linear systems, where the characteristic roots of each subsystem are a pair of complex conjugate roots with positive real parts, is investigated in this paper. For the switched systems with phase trajectories rotating in the same direction, a more appropriate state-dependent switching rule is formulated to guarantee system stability. In addition, a new switching rule is developed to stabilize the switched system in which the phase trajectories of two subsystems rotate in opposite directions.

Compared with previous findings, the proposed state-dependent switching rule has the advantages of fast convergence, simple computation and weak conservativeness. However, the limitation of the present method is that the considered switched system is overly simplistic. In view of this, we will try to extend the results to the case of higher-order subsystems in future, for which we need to find a matched mechanical model. Meanwhile, the proposed switching mechanism is expected in the case of multiple subsystems, in which the relationship between the switching sequence and subsystem characteristics warrant clarification. Both represent the focus of our future research work.

**Author Contributions:** The authors X.Y. and Y.Z. contributed equally to this work. All authors have read and agreed to the published version of the manuscript.

**Funding:** This work was supported by the National Natural Science Foundation under Grant (12162006), and the Cultivation Project of Guizhou University [2019] 63.

**Institutional Review Board Statement:** Not applicable.

**Informed Consent Statement:** Not applicable.

**Data Availability Statement:** Not applicable.

**Acknowledgments:** Not applicable.

**Conflicts of Interest:** The authors declare no conflict of interest.

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
