# Peer review of "The Design of State-Dependent Switching Rules for Second-Order Switched Linear Systems Revisited"

_axioms, doi:10.3390/axioms11100566_

Round 1

Reviewer 1 Report

The reviewed  manuscript devoted to study of the asymptotic stability of second-order switched linear systems  with positive real part conjugate complex roots for each subsystem. An  appropriate state-dependent switching rule is designed to stabilize a switched system with the  phase trajectories of two subsystems rotating outward in the same direction or the opposite direction. Finally, several numerical examples are used to illustrate  of the results.

I have some comments and recommendations:

1) The results are given for the case  N=2 and n=2. It will be interesting if  the authors  present the results for the case  N=3 and n=2. 

2) It would also be necessary to add example for the case N=3 and n=2.

3)  In conclusion, the authors should also note the possibility of generalization of the results for the case N > 3 and n>2.

In our opinion, the manuscript needs a major revision. 

Reviewer 2 Report

In this manuscript, the asymptotic stability issue of second-order switched linear systems was addressed, where there exist positive real part conjugate complex roots for each subsystem. A new state-dependent switching rule was designed to stabilize the switched system. Finally, two examples were given to illustrate the effectiveness of the

proposed method. Some comments for this manuscript are listed as follows:

1. In the introduction, I suggest that the author add more discussion on the recently developed switching rule method.

2. What are the distinct contribution of the proposed approach when compared with existing works?

3. The authors have made some minor grammatical mistakes in this article. Although it does not affect the overall reading, it is recommended to check and correct them carefully.

4. The authors only state what has been accomplished in the conclusion part. The advantages and limitations of the proposed strategy should be explained and meaningful future works should also be pointed out.

5. Some related works are provided as follows, which may be helpful for the improvement of this manuscript.

[a] Nonfragile consensus of nonlinear multiagent systems with intrinsic delays via aperiodic memory sampled-data control, International Journal of Robust and Nonlinear Control

[b] Performance Error Estimation and Elastic Integral Event Triggering Mechanism Design for TS Fuzzy Networked Control System Under DoS Attacks, IEEE Transactions on Fuzzy Systems

Round 2

Reviewer 1 Report

The revised version manuscript is included results of  the asymptotic stability of second-order switched linear systems with positive real part conjugate complex roots for each subsystem.  In the manuscript  are shown several numerical examples for illustrating  the effectiveness and superiority of the proposed method.

In the revised version manuscript the authors took into account  our recommendations and comments.